# Current Awareness Status of and Recommendations for Polycystic Ovarian Syndrome: A National Cross-Sectional Investigation of Central Jordan

**DOI:** 10.3390/ijerph20054018

**Published:** 2023-02-23

**Authors:** Nadia Muhaidat, Shahd Mansour, Majid Dardas, Jamil Qiqieh, Zeina Halasa, Leen Al-Huneidy, Jehad Samhouri, Rama Rayyan, Wahid AlOweiwi, Jamil AlMohtasib, Mohammad A. Alshrouf, Ghayda’a M. Al-Labadi, Layla H. Suboh, Abdallah Al-Ani

**Affiliations:** 1Department of Obstetrics & Gynecology, School of Medicine, The University of Jordan, Amman 11942, Jordan; 2School of Medicine, The University of Jordan, Amman 11942, Jordan; 3Department of Internal Medicine, King Hussein Medical Center, Amman 11855, Jordan; 4Jordan University Hospital, The University of Jordan, Amman 11942, Jordan; 5Office of Scientific Affairs and Research, King Hussein Cancer Center, Amman 11941, Jordan

**Keywords:** central Jordan, PCOS, recommendations, knowledge, awareness

## Abstract

Polycystic ovary syndrome (PCOS) is a common reproductive disorder that is related to a number of health issues and has an influence on a variety of metabolic processes. Despite its burden on the health of females, PCOS is significantly underdiagnosed, which is associated with lack of disease knowledge among females. Therefore, we aimed to gauge the awareness of PCOS in both the male and female population in Jordan. A descriptive cross-sectional study was conducted, targeting individuals over the age of 18 from Jordan’s central region. Participants were recruited through stratified random sampling. The questionnaire consisted of two domains, including demographics and knowledge of PCOS domains. A total of 1532 respondents participated in this study. The findings revealed that participants have overall adequate knowledge regarding PCOS’s risk factors, etiology, clinical presentation, and outcomes. However, participants demonstrated subpar familiarity of the association between PCOS and other comorbidities and the effect of genetics on PCOS. Women had more knowledge than men about PCOS (57.5 ± 6.06 vs. 54.1 ± 6.71, *p* = 0.019). In addition, older, employed, and higher-income populations showed significantly better knowledge than younger, unemployed, self-employed, and lower-income populations. In conclusion, we demonstrated that Jordanian women demonstrate an acceptable yet incomplete level of knowledge towards PCOS. We recommend establishing educational programs by specialists for the general population as well as medical personnel to spread accurate medical information and clarify common misconceptions about signs, symptoms, management, and treatment of PCOS, and nutritional knowledge.

## 1. Introduction

Polycystic ovary syndrome (PCOS) is the predominant endocrine disorder affecting women of reproductive age [1]. The Middle East is no exception, with 16% of its population being affected [2], and the impact of this complex syndrome is retained throughout the lifespan of patients as it forces them to engage in lifelong self-management [3]. This syndrome is a leading cause of infertility, and women with it have a higher miscarriage rate than other sub-fertile women. Studies estimated that one in seven women have PCOS, among which two out of three will not ovulate properly [4]. In addition, PCOS increases the risk of gynecological cancers such as endometrial, ovarian, and breast cancers, along with developing hypertension and type 2 diabetes [5]. Furthermore, it causes many physical (e.g., central obesity, acne, hirsutism, hair loss, and baldness) and psychological (e.g., depression, stress, and anxiety) symptoms, all of which have a significant negative impact on the patient’s life [6]. Nonetheless, not all cases will present with the entire spectrum of clinical symptoms [7].

Although still elusive, the literature agrees that PCOS has a highly complex etiology that involves the contribution of epigenetic changes and solid genetic components [8]. These factors alter the natural balance of hormones in females, causing many of the aforementioned symptoms. Affected hormones include gonadotrophin-releasing hormone, insulin, luteinizing/follicle-stimulating hormone, androgens, estrogen, growth hormone, and cortisol, among others [9]. Due to the enigmatic pathophysiology of PCOS and the variations found between cases, it is challenging to develop universal treatment for all patients. However, the symptoms can still be managed through different treatments and lifestyle changes [10].

According to the new Rotterdam criteria, PCOS is diagnosed if a minimum of two out of the three following conditions are met: polycystic ovarian morphology in an ultrasound assessment, anovulation, and/or androgen excess [11]. The primary diagnostic markers of PCOS are the absence or irregularities in the menstrual cycle, hirsutism, increased androgen production, and abnormal ovarian morphology [11]. Nonetheless, despite the rigorous and expanded diagnostic criteria, PCOS is underdiagnosed and understudied [12].

Women in Jordan have insufficient knowledge regarding PCOS and its complications, and the disease is under evaluated [13,14]. Therefore, there is a need to enhance women’s knowledge to avert severe consequences [14]. Furthermore, there is a need to assess the knowledge in other subgroups of society such as males as they are the primary decision makers in hierarchal societies similar to that of Jordan [15]. Hence the goal of this study was to investigate the predictors of PCOS awareness among the Jordanian populace.

## 2. Materials and Methods

### 2.1. Study Design, Setting, Duration, and Characteristics

A descriptive, cross-sectional study involving 1630 Jordanian individuals was conducted between January and February 2022. Participants over the age of 18, literate (i.e., able to read and write), and able to give voluntary consent were approached for recruitment. Individuals younger than 18, who refused to participate in the study, or had failed to complete at least 80% of the questionnaire were excluded. A total of 98 participants did not meet the study’s inclusion criteria, which left 1532 participants eligible for the final analysis. At an 80% power, 95% confidence interval, and a 5% margin of error, a minimum of 385 participants were needed to conduct statistical analysis of appropriate power. Participants were targeted through social media using a self-administered questionnaire developed on Google Forms. Utilized social media outlets included Facebook, WhatsApp, and Telegram. To detect statistical significance, the study utilized stratified random sampling to collect representative data from Central Jordan’s age, residential, and educational strata. The main study aim was to investigate the level of knowledge and predictors of PCOS awareness among the Jordanian population.

The Hashemite Kingdom of Jordan is an upper-middle-income country with a population of 10 million. The average per capita income was $4282.8 in 2021. Jordan comprises 12 administrative governorates belonging to one of three regions: North, Centre, and South. Approximately two-thirds of the Jordanian population reside in the Centre region. It is characterized by a rapidly expansive population pyramid with a life expectancy of around 75 years for both sexes.

### 2.2. Instrument Development

Our data were collected from Jordan’s central region through an electronic questionnaire, which was self-developed and consisted of questions from previously referenced literature [6,16,17]. The questionnaire was pilot tested on 22 participants for validity, internal consistency, and general logistical characteristics.

The questionnaire consisted of two domains including demographics and knowledge. The first domain comprised variables that could predict or may have impacted the recipient’s knowledge of PCOS. Questions pertaining to demographics were either dichotomous in nature (e.g., yes, no; male, female) or composed of mutually exclusive choices (e.g., single, married, widowed, divorced). The second knowledge-oriented domain consisted of 25 questions derived from aforementioned literature on PCOS knowledge and awareness and measured by a 5-point Likert scale (i.e., strongly disagree, disagree, neutral, agree, strongly agree). The domain demonstrated appropriate reliability with a Cronbach’s alpha of 0.834. Scoring was done by taking a cut-off value at the median. A score equal or below the median was considered poor knowledge, while a score above the median was considered adequate knowledge. 

### 2.3. Statistical Analysis

Data were analyzed using SPSS version 23. The data were described using variability analysis in the form of means ± standard deviation. The sociodemographic factors were calculated and provided as frequencies (percentages) using standard descriptive statistical parameters. For items in the PCOS knowledge domain utilizing 5-point Likert scales, disagreement responses were grouped together, while agreement responses were grouped together for ease in reporting. Associations between categorical variables were assessed using Chi-square. A spearman coefficient was calculated to test for correlation between categorical variables. Mean differences of every item within the questionnaire were compared between various categorical groups using the student’s t-test or ANOVA. Subsequently, a total knowledge score was determined by calculating the average mean of all items constituting said domain and compared between categories using the aforementioned statistical tests. A binary logistic regression model and a linear regression model were computed to explore predictors of PCOS knowledge. Multiple correction was conducted using the Holm–Bonferroni sequential method. Correction for regression analysis was not conducted as it was an exploratory analysis of controlled predictors. In addition, the conservative nature of correction reduces type I errors at the expense of increasing type II errors. Normal distribution of data was ensured by the Kolmogorov–Smirnov test of normality. Any variable with missing values of more than 80% of the total sample size was excluded to minimize selection bias. All statistical tests were conducted with 95% confidence interval and 5% error margin. A *p*-value of less than 0.05 was considered statistically significant.

## 3. Results

### 3.1. Demographic Characteristics

After excluding participants not meeting the study’s inclusion criteria, 94% (n = 1532) of respondents were eligible for the final analysis. The average age of respondents was 33.74 ± 12.45 years. The greater majority of participants were female (88.6%), lived in urban cities (92.0%), and were married (52.8%). In terms of education, 62.1% had a bachelor’s degree or higher and 57.8% of respondents had degrees not related to the medical field. Table 1 demonstrates the sociodemographic characteristics of the recruited sample.

### 3.2. Knowledge Regarding PCOS

Perceived knowledge of PCOS among recruited participants is demonstrated in Table 2. Most respondents (60.8%) did not know that PCOS patients do not necessarily have multiple ovarian cysts. Furthermore, PCOS’s association with heart disease, hypertension, and early puberty/menarche was not identified, even by respondents working in the medical field (refer to Appendix A). Respondents had adequate knowledge about all PCOS symptoms except unusual amounts of hair loss from the scalp, which was unknown or incorrectly answered by 63.9% of the respondents. On the other hand, 65% of the respondents correctly identified the correlation between PCOS and infertility. Lastly, the genetic component of PCOS was the least known aspect of the disease, with 32.9% of respondents answering incorrectly and 48.8% not knowing about it.

In addition, most participants were aware of losing weight (74.1%) and exercising (69.3%) as means to decrease the severity of PCOS symptoms. However, most participants were unaware of the effect of eating protein-rich food (27.5%) on relieving PCOS symptoms, and only a minority of participants knew that eating fat-rich food (6.1%) had a positive effect on PCOS symptoms (Figure 1).

### 3.3. Univariate Analysis

Our results demonstrate that participants whom were females (*p*-value = 0.02), single (*p*-value < 0.001), had higher educational levels (*p*-value < 0.001), and were working within a healthcare-related field (*p*-value < 0.001) had significantly higher PCOS knowledge scores. Additionally, there was a weak negative correlation between the age groups and the knowledge score (r = −1.41; *p*-value: <0.001). Upon post hoc analysis, females had significantly higher knowledge than males in the medical field and the general population (*p*-value = 0.019). Furthermore, there was no significant difference in knowledge scores between those with a bachelor’s degree and those with an even higher degree (*p*-value = 0.723). As for occupation, it was demonstrated that housewives had significantly lower knowledge scores than all the other occupations except those who were self-employed (*p*-value < 0.001). Lastly, there was a significant difference in knowledge scores between income groups (*p*-value = 0.005), with variances in PCOS knowledge between the 0–500 and more than 2000 income being the most evident (*p*-value = 0.004). Table 3 demonstrates the factors associated with PCOS knowledge.

### 3.4. Predictors of PCOS Knowledge (Regression Analysis)

Binary logistic regression demonstrated that older people, people with no education, and those who were in nonmedical study fields were more likely to have unsatisfactory knowledge of PCOS ((OR: 0.976; 95%CI: 0.962–0.989), (OR: 0.463; 95%CI: 0.263–0.816), (OR: 0.458; 95%CI: 0.343–0.613), respectively). On the linear regression model, similar predictions were found for old age, lack of education, and nonmedical study fields ((β: −0.086; 95%CI: −0.0810–−0.005), (β: −0.161; 95%CI: −4.810–−1.716), (β: −0.262; 95%CI: −4.086–−2.530), respectively). 

Similarly, the binary regression model showed that females and those with a monthly income over JOD 2000 were more likely to have appropriate PCOS knowledge ((OR: 1.281; 95%CI: 2.496–5.190) and (OR: 0.417; 95%CI: 1.038–2.217), respectively), while the linear regression model demonstrated that females, those who had a bachelor’s degree, those with higher educational levels, and those with a socioeconomic status over JOD 2000 were positive predictors of higher PCOS knowledge scores ((β: 0.205; 95%CI: 3.044–4.991), (β: 0.198; 95%CI: 0.627–4.468), (β: 0.167; 95%CI: 0.774–4.933) and (β: 0.066; 95%CI: 0.216–2.271), respectively). Table 4 shows the regression models.

## 4. Discussion

Our results demonstrated that participants had adequate knowledge in most categories except for the association between PCOS and other comorbidities and the effect of genetics on PCOS. As for demographics, location of residence was the only factor that did not have an effect on participants’ knowledge scores. Overall, females had significantly higher knowledge than males among all explored strata. Alternatively, housewives and self-employed participants portrayed poor PCOS knowledge.

A review of relevant literature in the region yielded some studies that aimed to assess knowledge and awareness of PCOS in individual countries. In the UAE, a convenience sample of 493 Emirati students was assessed for PCOS awareness. The study demonstrated low reproductive health knowledge and poor PCOS awareness, as only 38% of the female students selected were familiar with the condition [18]. A Jordanian study showed that female university students in Northern Jordan had satisfactory levels of PCOS awareness [19]. On the other hand, a survey-based study of 227 Jordanian women demonstrated insufficient knowledge about PCOS and its complications [14]. Lastly, a Saudi Arabian report on PCOS awareness revealed that over 60% of respondents had minimal awareness of PCOS [6]. The aforementioned studies displayed methodological heterogeneity, including using multiple data collection forms (i.e., online and paper), limiting the target population to students, using convenience sampling, using poorly validated data collection tools, and limiting participants’ ages to 50 years. Through the usage of a stratified sampling technique, online-only survey tools, and more inclusive selection criteria, our study managed to avoid some of the pitfalls of the current local and regional literature. Overall, in contrast to the aforementioned studies, our participants demonstrated adequate levels of PCOS knowledge and awareness.

Unique to most studies in the region, our study is the first to include males in the assessment of PCOS knowledge. Our results show that females have significantly more knowledge than their male counterparts. This perceived difference in knowledge could be attributed to lower levels of reproductive health literacy among men [20]. In Jordan and most developing countries, men are gatekeepers to health care [21]. They are the primary decision-makers and are in a position to directly affect their partner’s and children’s health [22]. Their decisions impact the distribution and utilization of resources, access to healthcare services, use of contraceptives, child spacing, availability of nutritious food, and women’s workload [23]. In addition, their actions, in terms of abuse or neglect, directly impact the health of their partners and children [24]. Due to the tremendous power held by men within the Jordanian family structure, which is both economic and social, the active involvement of men in maternal and child health (MCH) has significant implications.

The present study found a negative correlation between age and knowledge of PCOS, with people aged 60 and older having the lowest mean knowledge score. This can be attributed to the lower level of health literacy in the elderly [6]. Furthermore, physical and functional health problems, such as reduced vision, physical disability, and poor memory, can pose barriers to internet use and predispose older populations to technology biases [25].

The current study showed, as expected, that a higher educational level and an education in the medical field significantly increase participants’ knowledge about PCOS, as shown by a Saudi study [6]. This is important, as medical personnel are viewed as the primary source of information regarding reproductive issues [26] and, therefore, they should have more knowledge about PCOS [14,18,26]. Furthermore, housewives had the lowest level of knowledge compared with all the other occupations. Housewives and mothers must be aware of the signs and symptoms of PCOS to avoid overlooking the disease when their children complain to them about such symptoms [27].

The complexity of the disorder and its impact on quality of life require timely diagnosis, screening for complications, and management strategies. Unfortunately, PCOS remains underdiagnosed, and women experience significant delays in diagnosis [28]. Delays in diagnosis can lead to the progression of comorbidities, making it more challenging to implement lifestyle intervention, which is critical for the improvement of features of PCOS and quality of life [28]. Women with PCOS are at an increased risk of suffering from type 2 diabetes; metabolic syndrome; cardiovascular issues, such as hypertension and dyslipidemia; gynecological diseases, including infertility, endometrial dysplasia, endometrial cancer, and malignant ovarian tumors; pregnancy complications, such as premature birth, low birth weight, and eclampsia; and emotional and mental disorders in the future [29]. Moreover, PCOS accounts for more than 75% of cases of anovulatory infertility, which is frequently succeeded by impaired follicular maturation, anovulation, and biochemical pregnancy [29].

Housewives and mothers must also be aware of the genetic element of PCOS, as it is estimated that 20 to 40 percent of women with PCOS have an affected mother or sister [30]. This increased familial risk is partly due to shared genetic factors, but lifestyle influences shared by family members likely also play a role [31]. However, most participants were unaware of the etiology of PCOS and its genetic component. In a similar Saudi Arabian study, only 10.2% of participants were aware of the familial aspect of PCOS [6]. Another study reported that 25.9% of participants thought PCOS was due to genetic and acquired causes, while 33.3% did not know [26]. Even though the interaction between the environmental and genetic elements underlying PCOS has not been determined [32], a recent study concluded that DNA hypomethylation is a crucial epigenetic mechanism associated with regulating PCOS genes [33]. The experimental epigenetic-based therapy (S-adenosylmethionine) used in the aforementioned study was found to correct the transcriptomic, neuroendocrine, and metabolic defects in mice. These results might usher in a new era in the usage of individualized epigenetic drugs in the treatment/curettage of PCOS.

Certain lifestyle modifications, such as regularly exercising, losing weight, using oral contraceptives (OCs), and having a balanced diet rich in vegetables, fruits, proteins, and fats, are associated with improved PCOS symptoms [34,35]. Of these, patients identified the positive effects of exercising, losing weight, and selected aspects of a balanced diet, like eating vegetables and fruits. However, the association with having a diet rich in protein and fat was widely underappreciated, which may be due to the constitution of the Jordanian diet, where wheat and rice are considered the primary sources of daily dietary intake [36]. Furthermore, only 6.1% of respondents identified eating a fat-rich diet as a measure of alleviating PCOS symptoms, which may be partly due to the negative connotation of fats and the association between “fat” and cholesterol-rich foods high in trans fats rather than mono- and polyunsaturated fats that regulate a patient’s lipid profile [37]. In an Emirati study, 45% of respondents knew that PCOS could be managed with proper diet, exercise, and medication [18]. On the other hand, 11% of Northern European health professionals associated improvement of PCOS symptoms with a low glycemic index diet [38]. Overall, awareness of these measures was considerably higher when compared with a similar study carried out in Saudi Arabia [6], where losing weight was correctly identified by 45.3% of participants, as opposed to 74.1% of participants in this study.

The association with using OCs was less known to participants in this study, which necessitates education programs since, in Jordan, OCs are the second most popular method of contraception, preceded by intrauterine devices (IUD) [39]. In one study, it was concluded that Jordanian women had positive attitudes regarding the efficacy and safety of OCs. However, only half of women self-reported knowing how to use OCs [40]. In addition, no significant differences in OCs utilization patterns were found according to demographic variables (i.e., age, education, outcome, etc.) [40]. According to the 2012 Demographic Health Survey (DHS), around 42% of Jordanian married women are using a method of contraception, mainly the IUD (21%), followed by the hormonal contraceptive pill (8%), and male condom (8%) [39]. Furthermore, according to the Jordan Association for Family Planning and Protection, the price of an IUD insertion is 11.27 USD, and the price of one oral contraceptive pill course is 1.41 USD, which makes the options accessible to the Jordanian population [41].

One of the biggest misconceptions surrounding PCOS is that having atretic ovarian follicles on ultrasound imaging is a mandatory part of the Rotterdam criteria for diagnosing PCOS, whereas the diagnosis can be made without them [3]. This misconception was deeply rooted in the Jordanian population, as 60.8% of the participants agreed it was required for the diagnosis. This observation can also be supported by a previously released Jordanian paper, where 33% of participants thought the PCOS name was confusing and did not provide sufficient information regarding the disease [14].

Though the average percentage of knowledge of complications for Jordanians was 36.9%, compared with 23.7% in Saudi Arabia [6], knowledge of its effect on heart disease and hypertension was lacking (11% and 19.1%, respectively), even among those in the medical field. Despite having a low sample size, a limited age range, and only including the female sex, the previously published Jordanian study also supported our results that having heart disease was the least-known complication of PCOS [14]. On the other hand, 65% of the participants were aware of the psychological disturbances caused by PCOS, opposing what was discussed in the papers [6,38].

### 4.1. Limitations

This study was subject to a few limitations. First, the sample design only represented a snapshot of the population in Jordan and might be missing specific strata, especially from rural areas, which might reduce the generalizability of the results. Second, our data collection instrument was limited to people with a working internet connection, so anyone who could not access an online questionnaire could not participate in this study. Third, the study utilized a self-reported and close-ended data collection tool. Such a tool is prone to recall bias, extreme bias, or social desirability bias. Additionally, these questionnaires can be time consuming and may not be suitable for certain populations, such as those with low literacy or general language barriers. However, our strengths lay in recruiting a large sample comprising both sexes and representing the Jordanian population’s age distribution.

### 4.2. Recommendations

As agreed upon by many studies [14,18,26], primary care physicians (PCP) are considered the primary source of information regarding PCOS. However, since our results demonstrate a lack of medical personnel knowledgeable about the association between PCOS and other comorbidities, it is essential to enhance PCP knowledge about this aspect. Therefore, we propose conducting medical seminars hosted by specialist gynecologists and endocrinologists to spread accurate medical information among fellow medical personnel about the signs, symptoms, management, and treatment of PCOS. In addition, physicians treating PCOS patients must inform the diagnosed females of its hereditary nature and recommend having any symptomatic female relatives visit a physician.

As far as we know, there is no published Jordanian paper discussing the gap in nutritional knowledge among physicians, but other papers from neighboring and western countries do acknowledge that nutritional education among physicians is lacking [42,43,44,45]. Hence, we recommend that physicians be made more aware of how essential dietary and lifestyle modifications are in the management of PCOS in order to provide proper treatment for the condition and educate females about any alterations in their daily practices that can benefit the course of their treatment. The limited physician–patient time is another barrier to educating patients about the importance of nutrition in PCOS treatment [46]. Non-advanced practice nurses can help with this dilemma, as they are uniquely positioned to impact patient satisfaction and clinical outcomes by providing and reinforcing patient education, coordinating management processes, counseling via telephone or in-person appointments, and facilitating referrals to ancillary providers [47]. However, this recommendation entails the requirement of numerous nurses, which would further burden hospitals financially.

Furthermore, we recommend adding vitamin D3 supplements at a treatment dose of 50,000 IU per week to the national PCOS management plan, as it has proven to decrease the hirsutism scores, decrease the androgen levels, and increase the fertility of overweight women with PCOS in Jordan [48]. This recommendation can also be supported by the fact that Jordanian women have a higher risk of vitamin D3 deficiency, with 37.3% of the Jordanian female population being affected, compared with other females in western countries. This difference in risk for vitamin D3 deficiency has been attributed to the dressing styles of women in Jordan (e.g., Hijab and Nikab) [49].

Knowledge and awareness programs are also needed to inform the general population about PCOS and clarify misconceptions. These programs should primarily target housewives, mothers, and males (including those in the medical field) since those groups showed the least knowledge in our sample. We suggest launching these educational programs on accessible platforms such as social media, radio stations, podcasts, or educational television programs, where specialists can adequately educate the general population about PCOS. Furthermore, we can consider using private social networking (i.e., mobile applications and SMS) to promote PCOS awareness, as this method demonstrated a significantly better awareness outcome than a control group [50]. In another study, using an online video-based structured education module significantly increased PCOS awareness, especially among younger adults (age ≤ 26) [51].

## 5. Conclusions

The results of this study concluded that women had more knowledge than men about PCOS. In addition, older, employed, and higher-income populations showed significantly better knowledge than younger, unemployed, self-employed, and lower-income populations. However, everyone would benefit from further education, and therefore educational programs targeting these subgroups must be established.

## Figures and Tables

**Figure 1 ijerph-20-04018-f001:**
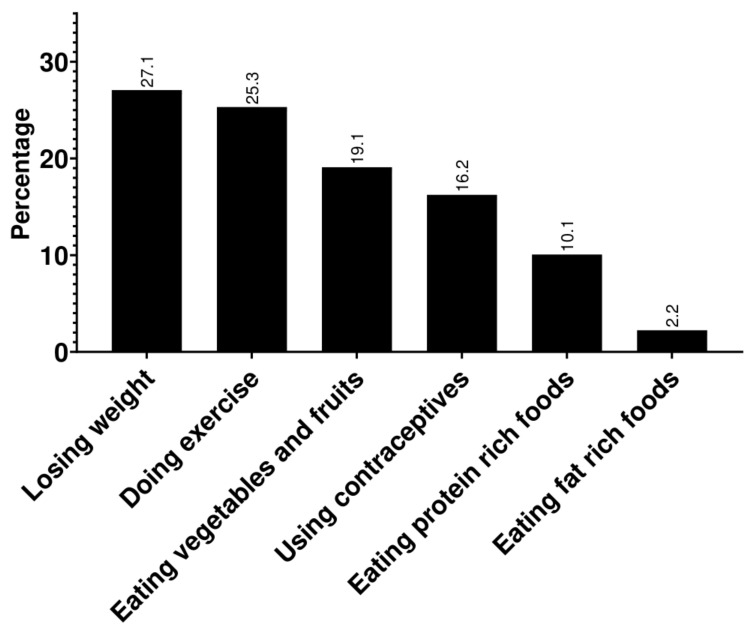
Knowledge of lifestyle modifications related to polycystic ovary syndrome symptoms relief.

**Table 1 ijerph-20-04018-t001:** Demographic characteristics of studied sample.

Variable		N	%
Age	18–29	664	43.3
	30–39	353	23.0
	40–49	333	21.7
	50–59	131	8.6
	60+	51	3.3
Sex	Male	175	11.4
	Female	1357	88.6
Marital status	Single	665	43.4
	Married	809	52.8
	Divorced/Widowed	58	3.8
Location	Urban	1410	92.0
	Rural	122	8.0
Education level	No certificate	67	4.4
	Highschool	273	17.8
	Bachelors	952	62.1
	Higher education	240	15.7
Field of study	No education	161	10.5
	Non-medical field	885	57.8
	Medical field	486	31.7
Occupation	Housewife	374	24.4
	Student	454	29.6
	Unemployed	67	4.4
	Employed	538	35.1
	Self-employed	99	6.5
Monthly income *	0–500	610	39.8
	500–1000	363	23.7
	1000–1500	253	16.5
	1500–2000	118	7.7
	2000+	188	12.3

* Monthly household income in JOD = Jordanian Dinar; 1 JOD = 1.41 USD.

**Table 2 ijerph-20-04018-t002:** Overall knowledge of PCOS among recruited participants.

Item	Response		
	Disagree	I Don’t Know	Agree
	n (%)	n (%)	n (%)
In PCOS, there is an increased level of androgen hormones (testosterone)	57 (3.7%)	967 (63.1%)	508 (33.2%)
All patients suffering from PCOS have multiple small cysts in their ovaries	600 (39.2%)	468 (30.5%)	464 (30.3%)
Obesity may cause PCOS	107 (7.0%)	322 (21.0%)	1103 (72.0%)
Prediabetes condition (due to decreased insulin action in body) may cause PCOS	125 (8.2%)	709 (46.3%)	698 (45.6%)
Irregular or absence of menstrual cycle (period) is a symptom of PCOS	31 (2.0%)	233 (15.2%)	1268 (82.8%)
Unusual amount of hair growth on different body parts (upper lip, chin, abdomen, breast, etc.) is a symptom of PCOS	63 (4.1%)	324 (21.1%)	1145 (74.7%)
Severe acne problems are a symptom of PCOS	180 (11.7%)	529 (34.5%)	823 (53.7%)
Unusual amounts of hair loss from the scalp is a symptom of PCOS	185 (12.1%)	795 (51.9%)	552 (36.0%)
PCOS diagnosis can be confirmed through an ultrasound	105 (6.9%)	430 (28.1%)	997 (65.1%)
Specific blood tests can be used for the diagnosis of PCOS	133 (8.7%)	591 (38.6%)	808 (52.7%)
PCOS may lead to diabetes	270 (17.6%)	810 (52.9%)	452 (29.5%)
PCOS may lead to weight gain	119 (7.8%)	398 (26.0%)	1015 (66.3%)
PCOS may lead to pelvic pain	87 (5.7%)	412 (26.9%)	1033 (67.4%)
PCOS may lead to heart diseases	377 (24.6%)	978 (63.8%)	177 (11.6%)
PCOS may lead to hypertension	303 (19.8%)	937 (61.2%)	292 (19.1%)
PCOS may lead to infertility (inability to have children)	147 (9.6%)	384 (25.1%)	1001 (65.3%)
PCOS may lead to pregnancy complications (ex: miscarriage)	124 (8.1%)	527 (34.4%)	881 (57.5%)
PCOS may lead to anxiety/depression/low self-esteem	78 (5.1%)	426 (27.8%)	1028 (67.1%)
PCOS may lead to early puberty/menarche	256 (16.7%)	955 (62.3%)	321 (21.0%)
Hormonal therapy may be used to treat PCOS	52 (3.4%)	546 (35.6%)	934 (61.0%)
Anti-diabetic medications (ex: Glucophage) may be used to treat PCOS	71 (4.6%)	659 (43.0%)	802 (52.3%)
Surgery may be used to remove the ovarian cysts	132 (8.6%)	397 (25.9%)	1003 (65.5%)
Treating PCOS reduces the chance of getting cancer	150 (9.8%)	813 (53.1%)	569 (37.1%)
PCOS is an inherited disorder	504 (32.9%)	748 (48.8%)	280 (18.3%)

**Table 3 ijerph-20-04018-t003:** Mean score of the knowledge score about PCOS and demographic characteristics.

Demographics		Knowledge Score	*p*-Value
Age			<0.001
	18–29	57.8 ± 6.3	
	30–39	56.8 ± 5.8	
	40–49	57.1 ± 5.9	
	50–59	55.7 ± 6.8	
	60+	52.0 ± 6.1	
Sex			0.019
	Male	54.1 ± 6.71	
	Female	57.5 ± 6.06	
Marital status			<0.001
	Single	56.1 ± 6.18	
	Married	54.7 ± 5.89	
	Widowed/Divorced	53.2 ± 6.66	
Location			0.209
	Urban	57.1 ± 6.18	
	Rural	56.0 ± 6.66	
Education level			<0.001
	No high school diploma	52.7 ± 5.93	
	Highschool diploma or equivalent	56.0 ± 6.36	
	Bachelors	57.6 ± 6.15	
	Higher education	57.4 ± 5.86	
Field of study			<0.001
	Medical	59.3 ± 6.02	
	Non-medical	56.4 ± 5.99	
	Has not completed their education	54.1 ± 6.02	
Occupation			<0.001
	Unemployed	58.5 ± 6.19	
	Housewife	55.7 ± 5.87	
	Student	58.0 ± 6.50	
	Self-employed	56.5 ± 7.16	
	Employed	57.1 ± 5.87	
Monthly income			<0.010
	0–500	56.6 ± 6.11	
	500–1000	57.1 ± 6.09	
	1000–1500	56.6 ± 6.36	
	1500–2000	58.4 ± 6.13	
	>2000	58.2 ± 6.23	

**Table 4 ijerph-20-04018-t004:** Predictors of PCOS knowledge.

	Knowledgeable about PCOS (YES/NO)				Knowledge SCORE			
	Binary Logistic Linear Regression				Linear Regression Model			
	*p*-Value	Odds Ratio	Lower 95% CI for (OR)	Upper 95% CI for (OR)	*p*-Value	Beta (β)	Lower 95% CI for (β)	Upper 95% CI for (β)
**Age**	**0.001**	0.976	0.962	0.989	**0.028**	−0.086	−0.0810	−0.005
**Sex ** (Male = Reference)	**<0.001**	3.599	2.496	5.190	**<0.001**	0.205	3.044	4.991
**Marital status ** (Reference = single)								
Married	0.208	1.260	0.880	1.803	0.545	0.024	−0.682	1.290
Divorced/widow	0.575	0.832	0.436	1.585	0.180	−0.038	−3.014	0.565
**Location ** (Reference = Urban)	0.312	1.235	0.820	1.862	0.296	−0.026	−1.724	0.525
**Education level ** (Reference = no certificate)								
Highschool certificate	0.291	1.432	0.736	2.789	0.063	0.102	−0.090	3.414
Bachelor’s degree	0.070	1.959	0.947	4.055	**0.009**	0.198	0.627	4.468
Higher education	0.063	2.103	0.960	4.610	**0.007**	0.167	0.774	4.933
**Field of study** (Reference = medical field)								
No education	**0.008**	0.463	0.263	0.816	**<0.001**	−0.161	−4.810	−1.716
Non-medical field	**<0.001**	0.458	0.343	0.613	**<0.001**	−0.262	−4.086	−2.530
**Occupation ** (Reference = self-employed)								
Housewife	0.840	1.054	0.636	1.746	0.475	−0.035	−1.902	0.886
Student	0.694	0.890	0.498	1.590	0.256	−0.068	−2.511	0.670
Unemployed	**0.024**	2.259	1.113	4.583	0.104	0.051	−0.325	3.460
Employed	0.130	1.435	0.899	2.289	0.537	0.031	−0.880	1.688
**Monthly income** (Reference = 0–500)								
500–1000	0.439	1.120	0.841	1.493	0.525	0.018	−0.539	1.055
1000–1500	0.784	0.955	0.685	1.331	0.399	−0.024	−1.324	0.528
1500–2000	0.093	1.477	0.937	2.326	0.120	0.041	−0.254	2.186
2000+	**0.031**	1.517	1.038	2.217	**0.018**	0.066	0.216	2.271

Note: Bold indicates statistical significance (*p* < 0.05), monthly household income in JOD = Jordanian Dinar; 1 JOD = 1.41 USD.

## Data Availability

Data will be provided at reasonable request from the corresponding author.

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
