# Peer review of "Current Awareness Status of and Recommendations for Polycystic Ovarian Syndrome: A National Cross-Sectional Investigation of Central Jordan"

_ijerph, 2023, doi:10.3390/ijerph20054018_

Round 1
Reviewer 1 Report
It is a very interesting work on a subject that causes not only clinical pictures "per se" but also sterility in many women.
The theoretical approach is consistent and adequate.
Just to mention, to the authors, some probable modifications:
They classify as gender (social construct) two items that only represent the sex of the respondents; in fact, sex is what is relevant in this pathology.
Perhaps the work would win if it were stratified by sex, since the knowledge of this pathology could be studied by each of the sexes, since the opinion of men can affect the results obtained. Possibly it would also be good to sub-stratify by educational level or occupation.
Table 2, seems too extensive, should be summarized and include the total in an annex.
If it is decided to stratify by sex, tables and results must be corrected.
It is a pity that the researchers have not worked on the sources from which the interviewees have obtained the information, given the results of the study. If they have this information, it would be relevant for the quality of the article to include some information about it.
Author Response
We thank the reviewers for their insightful comments and valid concerns. All points of constructive criticism pointed out by team were addressed and corrected accordingly (presented below). Your expertise enables us to produce quality work and for that we are extremely thankful. The entire manuscript was revised and reformatted in a way appropriate to the journal’s instructions and reviewer’s comments.
Reviewer 1:
- They classify as gender (social construct) two items that only represent the sex of the respondents; in fact, sex is what is relevant in this pathology.
Perhaps the work would win if it were stratified by sex, since the knowledge of this pathology could be studied by each of the sexes, since the opinion of men can affect the results obtained. Possibly it would also be good to sub-stratify by educational level or occupation.
Response: We thank the reviewer for their observation. It should be noted that within Middle Eastern communities, particularly those which are conservative and Muslim, both biological sex and gender are considered the same and these terms are used interchangeably. This is also a characteristic of the Arabic language, as the language does not have words differentiating between gender and biological sex. Responses are now stratified by gender/biological sex within the supplementary tables.
- Table 2, seems too extensive, should be summarized and include the total in an annex. If it is decided to stratify by sex, tables and results must be corrected.
Response: Indeed, table 2 will include only total responses, while responses stratified by gender/biological sex or field of study will be included within the supplementary material. This will limit the size of the table and provide readers with the maximum number of resources to understand all possible combinations of results. The results were altered accordingly to fit with these changes.
- It is a pity that the researchers have not worked on the sources from which the interviewees have obtained the information, given the results of the study. If they have this information, it would be relevant for the quality of the article to include some information about it.
Response: We thank the reviewer for their important suggestion, and we apologize for not including the interviewees’ source of information. In future studies, it would be valuable for us to further explore this aspect of the interviewees’ knowledge background, and we will keep this point in consideration.
Author Response
We thank the reviewers for their insightful comments and valid concerns. All points of constructive criticism pointed out by team were addressed and corrected accordingly (presented below). Your expertise enables us to produce quality work and for that we are extremely thankful. The entire manuscript was revised and reformatted in a way appropriate to the journal’s instructions and reviewer’s comments.
Reviewer 2:
- Aim of the current study should have been stated clearly in the abstract. Importance and significance of the study/literature gap is not presented.
Response: We thank the reviewer for their insightful comments and valid concerns. The following was added to demonstrate a clear aim for the study and the rationale for its conduction. The abstract now reads: “Polycystic ovary syndrome (PCOS) is a common reproductive disorder that is related to a number of health issues and has an influence on a variety of metabolic processes. Despite its burden on the health of females, PCOS is significantly underdiagnosed which is associated with lack of disease knowledge among female. Therefore, we aimed to gauge the awareness of PCOS in both the male and female population in Jordan.”
- This study investigates the knowledge of general population not physicians' one. please justify this statement
Response: We apologize for this confusing statement. The statement, while correct, it doesn’t fit in within the abstract and was removed.
- Clarify, knowledge of what?
Response: For clarification the new statement now reads: “The questionnaire consisted of two domains, including demographics and knowledge of PCOS domains.”
- Again, the readers understanding for this is dependent on the clarification requested in my previous comment. please clarify what are the categories mentioned here.
Response: For clarification, that part of the abstract now reads: “The findings revealed that participants have an overall adequate knowledge regarding PCOS’s risk factors, etiology, clinical presentation, and outcomes. However, participants demonstrated subpar familiarity for the association between PCOS and other comorbidities and the effect of genetics on PCOS.”
- The last sentence of the abstract should be a conclusion of whether the findings of the study achieve the study objectives and what should be recommended based on the findings.
Response: The conclusion was altered as to fix smoothly with the abstract. The conclusion now reads “In conclusion, we demonstrated that Jordanian women demonstrate an acceptable yet incomplete level of knowledge towards PCOS. We recommend establishing educational programs by specialists for the general population as well as medical personnel to spread accurate medical in-formation and clarify common misconceptions about signs, symptoms, management, treatment of PCOS, and nutritional knowledge.”
- I would recommend adding PCOS
Response: “PCOS” was added to the list of keywords.
- elaborate more about the clinical impact of this disorder (endocrine, metabolic..etc)
Response: We greatly appreciate the reviewer’s input on our introduction. Therefore, it was restructured in order to satisfy the reviewer’s suggestions.
- menstruation disorders and its impact on women fertility is another major concern. please clarify this more.
Response: We thank the review for his recommendation. The following was added: “This syndrome is a leading cause of infertility, and women with it have a higher miscarriage rate than other sub-fertile women. Studies estimated that one in seven women have PCOS, among which two out of three will not ovulate properly [4].”
- which hormones?
Response: The following was added to delineate which hormones were affected: “These factors alter the natural balance of hormones in females, causing many of the aforementioned symptoms. Affected hormones include gonadotrophin-releasing hormone, insulin, luteinizing/follicle-stimulating hormone, androgens, strong, growth hormone, cortisol among others [9].”
- describe these symptoms briefly & list a reference here
Response: All symptoms relating to PCOS were moved to the first paragraph in our effort to restructure the manuscript.
- Move this above where elaboration on symptom was recommended
Response: Refer to Response to Comment 10.
- support this claim with reference(s)
Response: We thank the reviewer for his attention. Two references were added to support the claim.
- This is an interesting point. Why it is important to asses the knowledge of men about PCOS
Response: We thank the reviewer for his feedback. Middle Eastern communities such as Jordan, are patriarchal in nature and hierarchical in design. Men within these societies are the primary decision makers. These decisions extend to the health status and wellbeing of females. Therefore, it was interesting to gauge PCOS knowledge in men. The following was added: “Furthermore, there is a need to assess the knowledge in other subgroups of society such as males as they are the primary decision makers in hierarchal societies similar to that of Jordan [15].”
- I found the following methodological aspects deficient:
- Sample size estimation and power analysis.
- How many people were invited to participate in the study? How many were excluded? What were the reason(s) for exclusion? Were there any missing data? How missing data were treated- what type of imputation method was used? What was the response rate? All these data/ information were not described.
- The authors have not describe how the questions were phrased for assessment of demographic information and PCOS knowldge (for example yes/no questions, multiple choice questions, Likert- scale questions types...etc)
Response: For the aforementioned comments. The sample size calculation is now merged with the methodology’s first paragraph. At an 80% power, 95% confidence interval, and a 5% margin of error, a minimum of 385 participants were needed to conduct statistical analysis of appropriate power for a population of 10.4 million. These numbers are only accurate if we randomly sample participants.
As for the 2nd comment, our study sampled the entire population of Central Jordan. Therefore, we used G*Power to calculate the minimum number of participants in order to conduct relevant statistical testing representative of the overall population (as mentioned above). Since there is no finite amount of target population, a response rate cannot be calculated. Moreover, participants with missing data were excluded. Missing data up to 10% was tolerated within each variable. Reason for exclusion was mainly incomplete questionnaire responses.
As for the third comment, the wording of the questionnaire’s questions is now provided: “Questions pertaining to demographics were either dichotomous in nature (e.g., yes, no; male, female) or composed of mutually exclusive choices (e.g., single, married, widowed, divorced) The second knowledge-oriented domain consisted of 25 questions derived from aforementioned literature on PCOS knowledge and awareness and are measured by a 5-point Likert scale (i.e., strongly disagree, disagree, neutral, agree, strongly agree).”
- This word has multiple meanings. I think you mean "can read and write).
Response: “(can read and write)” was added after the word literate to communicate an accurate meaning
- example on grammatical issue
Response: Grammatical issue was resolved by replacing “given” with “give”
- This paragraph has irrelevant information except for the population size. however, the population size was not recruited to calculate the required sample size, which is missing.
Response: We might have to disagree with the esteemed reviewer. Firstly, the population size was utilized for sample size calculation as mentioned in response to comment 14. Secondly, due to the international readership of IJERPH, as introductory statement of Jordan’s vital statistics is needed. This includes the population structure/pyramid, income level, and overall structure of the country as the sampling was based only in Central Jordan. This gives international readers and future researchers a background on which they can contextualize and understand our results.
- I found this statement irrelevant here. what is the importance of mentioning the average income for the Jordanian individuals here?
Response: In addition to what’s above, demonstrating a measure of income is important as one of our recommendations deal with OCs and their prices. Moreover, a measure of income is needed to assert whether our recommendations fit a low-to-middle income setting or not.
- Same comment here. What is the importance of average life style in the methodology section? Please list these references
Response: Refer to response to comment number 17
- Please see my comment about the sample size calculation and rearrange the methodology section accordingly
Response: Please refer to response to comment number 14
- Please refer to my comment above about phrasing of questionnaire questions.
Response: Please refer to response to comment number 14
- The findings of the study described in the text were duplication of what were already mentioned in the tables. In fact, the authors should summarize the findings and include only the salient features of the study rather than repeating what already presented in the tables.
Response: We appreciate the reviewer’s concern, yet, we don’t grasp its full implications. The in-text results section contained only the significant and most pertinent observations. A myriad of other results was kept only in the tables. More analysis is now present as supplementary material. We choose items with statistical significance to be our salient and most important results. Nonetheless, sections describing table 2 and figure 1 were summarized to reduce redundancy.
- This should be used earlier in the text. for example it was mentioned in line 273-274.
Response: The abbreviation form of oral contraceptives (OCs) was transferred to the first time it was mentioned.
- Support this statement with a reference.
Response: A reference was added supporting the statement.
- Discuss the limitation of self-reported questionnaire.
Response: An entire section was added delineating the limitations of self-reported questionnaires. The section reads: “Third, the study utilized a self-reported and close-ended data collection tool. Such a tool is prone to recall bias, extreme bias, or social desirability bias. Additionally, these questionnaires can be time consuming and may not be suitable for certain populations, such as those with low literacy or general language barriers.”
Round 2
Reviewer 2 Report
looks satisfactory.